# Learning to Teach

**Yang Fan**[†,*] **Fei Tian**[‡,*]**, Tao Qin**[‡]**, Xiang-Yang Li**[†] **& Tie-Yan Liu**[‡]
[†]School of Computer Science and Technology, University of Science and Technology of China
`fyabc@mail.ustc.edu.cn, xiangyangli@ustc.edu.cn`
[‡]Microsoft Research
`{fetia,taoqin,tyliu}@microsoft.com`

## Abstract

Teaching plays a very important role in our society, by spreading human knowledge and educating our next generations. A good teacher will select appropriate teaching materials, impact suitable methodologies, and set up targeted examinations, according to the learning behaviors of the students. In the field of artificial intelligence, however, one has not fully explored the role of teaching, and pays most attention to machine *learning*. In this paper, we argue that equal attention, if not more, should be paid to teaching, and furthermore, an optimization framework (instead of heuristics) should be used to obtain good teaching strategies. We call this approach "learning to teach". In the approach, two intelligent agents interact with each other: a student model (which corresponds to the learner in traditional machine learning algorithms), and a teacher model (which determines the appropriate data, loss function, and hypothesis space to facilitate the training of the student model). The teacher model leverages the feedback from the student model to optimize its own teaching strategies by means of reinforcement learning, so as to achieve teacher-student co-evolution. To demonstrate the practical value of our proposed approach, we take the training of deep neural networks (DNN) as an example, and show that by using the learning to teach techniques, we are able to use much less training data and fewer iterations to achieve almost the same accuracy for different kinds of DNN models (e.g., multi-layer perceptron, convolutional neural networks and recurrent neural networks) under various machine learning tasks (e.g., image classification and text understanding).

## 1 Introduction

The evolution of modern human society heavily depends on its advanced education system. The goal of education is to equip the students with necessary knowledge and skills, so as to empower them to further deepen the understanding of the world, and push the frontier of our humanity. In general, the growth of a student will be influenced by two factors: his/her own learning ability and the teaching ability of his/her teacher. Among these two, the teacher plays a critical role: an experienced teacher enables faster learning of a student through elaborated strategies such as selecting appropriate teaching materials, imparting suitable methodologies, and setting up targeted examinations.

The training of an agent in artificial intelligence (e.g., an image classification model) is very similar to the growth of a student in human society. However, after carefully revisiting the literature of artificial intelligence (AI), we find that the importance role of the *teacher* has not been fully realized. Researchers put most of their efforts on the *student*, e.g., designing various optimization algorithms to enhance the learning ability of intelligent agents. In contrast, there are very limited attempts on building good teaching strategies, as briefly summarized below. Machine teaching (Zhu, 2013; 2015; Liu & Zhu, 2016; Liu et al., 2017) studies the problem of how to identify the smallest training set to push the machine learning model towards a pre-defined oracle model. Curriculum learning (CL) (Bengio et al., 2009; Spitkovsky et al., 2010; Graves et al., 2017) and self-paced learning (SPL) (Kumar et al., 2010; Lee & Grauman, 2011; Jiang et al., 2014b) heuristically define the scheduling of training data in a from-easy-to-hard order. Graduated optimization (Hazan et al., 2016) heuristically

---

[*]The first two authors contribute equally to this paper. Works done when the first author was visiting Microsoft Research Asia.

refines the non-convex loss function in a from-smooth-to-sharp manner, in order to make the machine learning process more robust. These attempts are either based on task-specific heuristic rules, or the strong assumption of a pre-known oracle model. In this regard, these works have not reflected the nature of education and the best practices in human society, where a good teacher is able to adaptively adopt different teaching strategies for different students under different circumstances, and is good at constantly improving his/her own teaching skills based on the feedback from the students.

In this paper, we argue that a formal study on the role of 'teaching' in artificial intelligence is sorely needed. Actually, there could be a natural analogy between teaching in artificial intelligence and teaching in human society. For example, selecting training data corresponds to choosing right teaching materials (e.g. textbooks); designing the loss functions corresponds to setting up targeted examinations; defining the hypothesis space corresponds to imparting the proper methodologies. Furthermore, an optimization framework (instead of heuristics) should be used to update the teaching skills based on the feedback from the students, so as to achieve teacher-student co-evolution. Just as French essayist Joseph Joubert said – "To teach is to learn twice", we call this new approach "learning to teach" (L2T).

In the L2T framework, there are two intelligent agents: a student model/agent, corresponding to the learner in traditional machine learning algorithms, and a teacher model/agent, determining the appropriate data, loss function, and hypothesis space to facilitate the learning of the student model. The training phase of L2T contains several episodes of sequential interactions between the teacher model and the student model. Based on the state information in each step, the teacher model updates the teaching actions so as to refine the machine learning problem of the student model. The student model then performs its learning process based on the inputs from the teacher model, and provides reward signals (e.g., the accuracy on the held-out development set) back to the teacher afterwards. The teacher model then utilizes such rewards to update its parameters via policy gradient methods (e.g., REINFORCE (Williams, 1992)). This interactive process is end-to-end trainable, exempt from the limitations of human-defined heuristics. Once converged, the teacher model could be applied to new learning scenarios and even new students, without extra efforts on re-training.

To demonstrate the practical value of our proposed approach, we take a specific problem, training data scheduling, as an example. We show that by using our method to adaptively select the most suitable training data, we can significantly improve the accuracy and convergence speed of various neural networks including multi-layer perceptron (MLP), convolutional neural networks (CNNs) and recurrent neural networks (RNNs), for different applications including image classification and text understanding. Furthermore, the teacher model obtained by our method from one task can be smoothly transferred to other tasks. For example, with the teacher model trained on MNIST with the MLP learner, one can achieve a satisfactory performance on CIFAR-10 only using roughly half of the training data to train a ResNet model as the student.

## 2 RELATED WORK

Our work connects two recently emerged trends of machine learning.

First, machine learning has evolved from simple learning to advanced learning. Representative works include learning to learn (Schmidhuber, 1987; Thrun & Pratt, 2012), or meta learning, which explores the possibility of automatic learning via transferring generic knowledge learnt from meta tasks. The two-level setup including meta-level model evolves slowly and task-level model progresses quickly is regarded to be important in improving AI. Recently meta learning has been widely adopted in quite a few machine learning scenarios. Several researchers try to design general optimizers or neural network architectures based on meta learning (Hochreiter et al., 2001; Andrychowicz et al., 2016; Li & Malik, 2016; Zoph & Le, 2017). Meta learning has also been studied in few-shot learning scenarios (Santoro et al., 2016; Munkhdalai & Yu, 2017; Finn et al., 2017).

Second, teaching has gradually attracted attention from researchers and been evolved as a new research direction in recent years from its origin several decades ago (Anderson et al., 1985; Goldman & Kearns, 1995). The recent efforts on teaching can be classified into two categories: machine-teaching and hardness based methods. The goal of machine teaching (Zhu, 2015; 2013) is to construct a minimal training set for the student model to learn a target model (i.e., an oracle). (Liu & Zhu, 2016)) define the teaching dimension of several learners. (Liu et al., 2017) extend ma-

chine teaching from batch settings to iterative setting. But with the strong assumption of oracle existence, machine teaching is applied in limited areas such as security (Scott Alfeld, 2017) and human-computer interaction (Suh et al., 2016). Without the assumption of the existence of the oracle model, hardness based methods assume that a data order from easy instances to hard ones benefits learning process. The measure of hardness in curriculum learning (CL) (Bengio et al., 2009; Spitkovsky et al., 2010; Tsvetkov et al., 2016; Graves et al., 2017) is typically determined by heuristic understandings of data. As a comparison, self-paced learning (SPL) (Kumar et al., 2010; Lee & Grauman, 2011; Jiang et al., 2014a;b; Supancic & Ramanan, 2013) quantifies the hardness by the loss on data. There are parallel related work (Graves et al., 2017) exploring several reward signals for automatically adapting data distributions along LSTM training. The teaching strategies in (Graves et al., 2017) are on per-task basis without any generalization ability to other learners. Furthermore, another literature called 'pedagogical teaching' (Shafto et al., 2014), especially its application to inverse reinforcement learning (IRL) (Ho et al., 2016) is much closer to our setting in that the teacher adjusts its behavior in order to facilitate student learning, by communicating with the student (i.e., showing not doing). However, apart from some differences in experimental setup and application scenarios, the applications of pedagogical teaching in IRL implies that the teacher model is still much stronger than the student, similar to the oracle existence assumption since there is an expert in IRL that gives the (state, action) trajectories based on the optimal policy.

The above works related to teaching have certain limitations. First, while a learning problem (e.g., the mathematic definition of binary classification (Mohri et al., 2012)) has been formally defined and studied, the teaching problem is not formally defined and thus it is difficult to differentiate a teaching problem from a learning problem. Second, most works rely on heuristic and fixed rules for teaching, which are task specific and not easy to apply to general teaching tasks.

## 3 LEARNING TO TEACH

In this section, we will formally define the framework of learning to teach. For simplicity and without loss of generality, we consider the setting of supervised learning in this section.

### 3.1 PROBLEM DEFINITION

In supervised learning, we are given an input (feature) space $\mathcal{X}$ and an output (label) space $\mathcal{Y}$; for any sample $x$ drawn from the input space according to a fixed but unknown distribution $P(x)$, a supervisor returns a label $y$ according to a fixed but unknown conditional distribution $P(y|x)$; the goal of supervised learning is to choose a function $f_\omega(x)$ with parameter vector $\omega$ that can predict the supervisor's label in the best possible way. The goodness of a function $f$ with parameter $\omega$ is evaluated by the risk

$$R(\omega) = \int \mathcal{M}(y, f_\omega(x)) \, dP(x, y),$$

where $\mathcal{M}(,)$ is the metric to evaluate the gap between the label and the prediction of the function.

One needs to consider several practical issues when training a machine learning model. First, as the joint distribution $P(x, y) = P(x)P(y|x)$ is unknown, the selection of a good function $f$ is based on a set of training data $D = \{x_i, y_i\}_{i=1}^n$. Second, since the metric $\mathcal{M}(,)$ is usually discrete and difficult to optimize, in training one usually employs a surrogate loss $L$. Third, to search for a good function $f$, a space of hypothesis functions should be given in advance, and one uses $\Omega$ to denote the set of parameters corresponding to the hypothesis space. Thus, the training process actually corresponds to the following optimization problem:

$$\omega^* = \arg\min_{\omega \in \Omega} \sum_{(x,y) \in D} L(y, f_\omega(x)) \triangleq \mu(D, L, \Omega). \tag{1}$$

As a summary, in conventional machine learning, a learning algorithm takes the set of training data $D$, the function class specified by $\Omega$, and the loss function $L$ as inputs, and outputs a function with parameter $\omega^*$ by minimizing the empirical risk $\min_{\omega \in \Omega} \sum_{(x,y) \in D} L(y, f_\omega(x))$. We use $\mu()$ to denote a learning algorithm, and we call it the *student* model to differentiate from the teaching algorithm defined as below.

In contrast to traditional machine learning, which is only concerned with the student model, in the learning to teach framework, we are also concerned with a teacher model, which tries to provide appropriate inputs to the student model so that it can achieve low risk functional $R(\omega)$ as efficiently as possible:

- **Training data**. The teacher model outputs a good training set $D \in \mathcal{D}$ to facilitate the training of the student model, where $\mathcal{D}$ is the Borel set on $(\mathcal{X}, \mathcal{Y})$ (i.e., the set of all possible training set). Data plays a similar role to the teaching materials such as textbooks in human teaching.

- **Loss function**. The teacher model designs a good loss function $L \in \mathcal{L}$ to guide the training process of the student model, where $\mathcal{L}$ is the set of all possible loss functions. As an analogy, the loss corresponds to the examination criteria for the student in human teaching.

- **Hypothesis space**. The teacher model defines a good function class $\Omega \in \mathcal{W}$, such as linear function class and polynomial function class, for the student model to search from, where $\mathcal{W}$ is the set of all possible hypothesis spaces. This also has a good analogy in human teaching: in order to solve a mathematical problem, middle school students are only taught with basic algebraic skills whereas undergraduate students are taught with calculus. The choice of different hypothesis spaces $\Omega$ will lead to different optimization difficulty, approximation errors, and generalization errors (Mohri et al., 2012).

The goal of the teacher model is to provide $D$, $L$ and $\Omega$ (or any combination of them) to the student model such that the student model either achieves lower risk $R(\omega)$ or progresses as fast as possible. Taking the first case as an example, the goal of the teacher model, denoted as $\phi$, is:

$$\min_{D,L,\Omega} \mathcal{M}(\mu(D, L, \Omega), D_{test}). \tag{2}$$

For ease of reference, we use $\mathcal{A}$ to represent the output space of the teacher model. It can be any combination of $\mathcal{D}$, $\mathcal{L}$ and $\mathcal{W}$. When $\mathcal{A}$ only contains $\mathcal{D}$, we call the special case "data teaching".

### 3.2 FRAMEWORK

As reviewed in Section 2, existing works that also consider the teaching strategies simply employ some heuristic rules and are task specific. In this subsection, we propose to model the learning and teaching strategies in L2T as a sequential decision process, as elaborated below.

- $S$ is a set of states. The state $s_t \in S$ at each time step $t$ represents the information available to the teacher model. $s_t$ is typically constructed from the current student model $f_{t-1}$ and the past teaching history of the teacher model.

- At the $t$-th step, given the state $s_t$, the teacher model takes an action $a_t \in \mathcal{A}$. Depending on specific teaching tasks, $a_t$ can be (1) a set of training data, (2) a loss function, or (3) a hypothesis space.

- $\phi_\theta : S \to \mathcal{A}$ is the policy with parameter $\theta$ employed by the teacher model to generate its action: $\phi_\theta(s_t) = a_t$. When without confusion, we also call $\phi_\theta$ the teacher model.

- The student model takes $a_t$ as input and outputs a function $f_t$, by using conventional machine learning technologies.

During the training phase of the teacher model, the teacher model keeps interacting with the student model. In particular, it provides the student model with a subset $\mathcal{A}_{train}$ from $\mathcal{A}$ and takes the performance of the learned student model as a feedback to update its own parameter. After the convergence of the training process, the teacher model can be used to teach either new student models, or the same student models in new learning scenarios such as another subset $\mathcal{A}_{test}$ is provided. Such a gener-

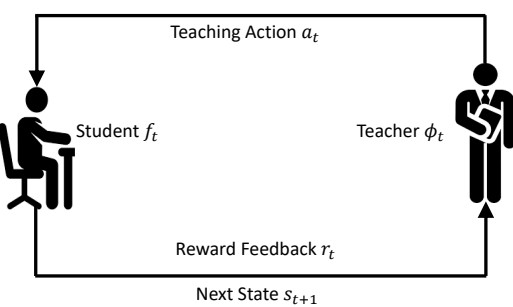

4

Figure 1: The interactive process between teacher and learner.

alization is feasible as long as the state representations $S$ are the same across different student models and different scenarios. As an example case, in the case of *data teaching* where $\mathcal{A} = \mathcal{D}$, in the training process teacher model $\phi_\theta$ could be optimized via the interaction with an MLP learner by selecting data from the MNIST dataset (acted as $\mathcal{A}_{train}$), and then the learned teacher model can be applied to teach a CNN student model on the CIFAR-10 dataset (acted as $\mathcal{A}_{test}$).

While one can choose different approaches to train the teacher model, in this paper, we employ reinforcement learning (RL) for this purpose. In this case, the teacher model $\phi_\theta$ acts as the *policy* interacting with the *environment*, which is represented by $S$. After seeing the teaching *action* $a_t$, the student model updates itself based on $a_t$, changes the environment to $s_{t+1}$ and then provides a *reward* $r_t$ to the teacher model. The reward indicates how good the current student model $f_t$ is, e.g., measured by the evaluation measure $\mathcal{M}$ on a held-out validation set. The teacher model then updates its own parameters in $\phi_\theta$ to maximize the accumulated reward. Such an interactive process between the teacher model and the student model is illustrated in Fig. 1. The interaction process stops when the student model get converged, forming one *episode* of the teacher model training.

Mathematically speaking, taking *data teaching* as an example in which $L$ and $\Omega$ are fixed, the objective of the teacher model in the L2T framework is:

$$\max_\theta \sum_t r_t = \max_\theta \sum_t r(f_t) = \max_\theta \sum_t r(\mu(\phi_\theta(s_t), L, \Omega)), \tag{3}$$

where $s_t$ is the $t$-th step state in the interaction of student model $\mu$ and teacher model $\phi$.

## 4 APPLICATION TO DATA TEACHING FOR NEURAL NETWORKS

In this section, taking data scheduling as an example, we show how to fully leverage the proposed learning to teach framework to help deep neural network training.[1]

### 4.1 STUDENT AND TEACHER SETUP

The student model $f$ is the deep neural network model for several real-world classification tasks. The evaluation measure $\mathcal{M}$ is therefore the accuracy. The student model obeys mini-batch stochastic gradient descent (SGD) as its learning rule (i.e., the $\arg\min$ part in Eqn. 1). Mini-batch SGD is a sequential process, in which mini-batches of data $\{D_1, \cdots D_t, \dots\}$ arrive sequentially in a random order. Here $D_t = (d_1, \cdots, d_M)$ is the mini-batch of data arriving at the $t$-th time step and consisting of $M$ training instances. The teacher model is responsible to provide training data to the student, i.e, $\mathcal{A} = \mathcal{D}$. Considering the sequential nature of SGD, essentially the teacher model wants to actively determine what is the next mini-batch data $D_t$ for the student. Furthermore, in reality it is computationally prohibitive to scan over all the remaining training data to select out $D_t$ at each step. To overcome this, after receiving the randomly arrived mini-batch $D_t$ of $M$ training instances, our teacher model $A$ dynamically determine which instances in $D_t$ are used for training and the others are abandoned. By teaching with appropriate data, the teacher aims to help the student model $f$ make faster progress, as reflected by the rapid improvement of $\mathcal{M}(f, D_{test})$.

### 4.2 MODELLING THE INTERACTION OF TEACHER AND STUDENT VIA REINFORCEMENT LEARNING

We introduce in details on how to leverage reinforcement learning to model the interaction between student and teacher. That is, the concrete concepts for $s_t$, $a_t$ and $r_t$ introduced in Subsection 3.2. For the state representation $S$, it corresponds to the mini-batch data arrived and current state of the deep neural network (i.e., the student): $s_t = (D_t, f_t)$. The teacher's actions are denoted via $a = \{a_m\}_{m=1}^M \in \{0, 1\}^M$, where $M$ is the batch size and $a_m \in \{1, 0\}$ denotes whether to keep the

---

[1]The experiments on teaching for other scenarios such as choosing $L$ and $F$ are easy to conduct as long as the teaching domain $\mathcal{A}$ is similarly defined.

$m$-th data instance in $D_t$ or not[2]. Those filtered instances will have no effects to student training. To encourage fast teaching convergence, we set the reward to be related with how *fast* the student model learns. Concretely speaking, $r$ is set as the terminal reward, with $r_t = 0, \forall t < T$, and $r_T$ is computed in the following way: we set an accuracy threshold $\tau \in [0, 1]$ and record the first mini-batch index $i_\tau$ in which the accuracy on a held-out dev set $D'_{dev}$ exceeds $\tau$, then set $r_T$ as $r_T = -\log(i_\tau/T')$, where $T'$ is a pre-defined maximum iteration number.

The teacher model sample its action $a_t$ per step by its policy $\phi_\theta(a|s)$ with parameters $\theta$ to be learnt. The policy $\phi_\theta$ can be any binary classification model, such as logistic regression and deep neural network. For example, $\phi_\theta(a|s) = a\sigma(w \cdot g(s) + b) + (1 - a)(1 - \sigma(\theta g(s) + b))$, where $\sigma(\cdot)$ is the sigmoid function, $\theta = \{w, b\}$ and $g(s)$ is the feature vector to effectively represent state $s$, discussed as below.

**State Features**: The aim of designing state feature vector $g(s)$ is to effectively and efficiently represent state $s$ (Graves et al., 2017). Since state $s$ includes both arrived training data and student model, we adopt three categories features to compose $g(s)$:

- Data features, contain information for data instance, such as its label category (we use 1 *of* $|Y|$ representations), (for texts) the length of sentence, linguistic features for text segments (Tsvetkov et al., 2016), or (for images) gradients histogram features (Dalal & Triggs, 2005). Such data features are commonly used in curriculum learning (Bengio et al., 2009; Tsvetkov et al., 2016).

- Student model features, include the signals reflecting how *well* current neural network is trained. We collect several simple features, such as passed mini-batch number (i.e., iteration), the average historical training loss and historical validation accuracy. They are proven to be effective enough to represent the status of current student model.

- Features to represent the combination of both data and learner model. By using these features, we target to represent how *important* the arrived training data is for current leaner. We mainly use three parts of such signals in our classification tasks: 1) the predicted probabilities of each class; 2) the loss value on that data, which appears frequently in self-paced learning (Kumar et al., 2010; Jiang et al., 2014a; Sachan & Xing, 2016); 3) the margin value.

The state features $g(s)$ are computed after the arrival of each mini-batch of training data. For a concrete feature list, as well as an analysis of different importance of each set of features, the readers may further refer to Appendix Subsection 8.3.

## 4.3 Optimization By Policy Gradient

The teacher model is trained by maximizing the expected reward: $J(\theta) = E_{\phi_\theta(a|s)}[R(s, a)]$, where $R(s, a)$ is the state-action value function. Since $R(s, a)$ is non-differentiable w.r.t. $\theta$, we use RE-INFORCE (Williams, 1992), a likelihood ratio policy gradient algorithm to optimize $J(\theta)$ based on the gradient: $\nabla_\theta = \sum_{t=1}^{T} E_{\phi_\theta(a_t|s_t)}[\nabla_\theta \log \phi_\theta(a_t|s_t)R(s_t, a_t)]$, which is empirically estimated as $\nabla_\theta \approx \sum_{t=1}^{T} \nabla_\theta \log \phi_\theta(a_t|s_t)v_t$. Here $v_t$ is the sampled estimation of reward $R(s_t, a_t)$ from one episode execution of the teaching policy $\phi_\theta(a|s)$. Given the reward is terminal reward, we finally have $\nabla_\theta \approx \sum_{t=1}^{T} \nabla_\theta \log \phi_\theta(a_t|s_t)r_T$.

---

[2]We consider data instances within the same mini-batch are independent with each other, and therefore for statement simplicity, when the context is clear, $a$ will be used to denote the remain/filter decision for single data instance, i.e., $a \in \{1, 0\}$. Similarly, the notation $s$ will sometimes represent the state for only one training instance.

## 5 EXPERIMENTS

### 5.1 EXPERIMENTS SETUP

#### 5.1.1 TASKS AND STUDENT MODELS

We conduct comprehensive experiments to test the effectiveness of the L2T framework: we consider three most widely used neural network architectures as the student models: multi-layer perceptron (MLP), convolutional neural networks (CNNs) and recurrent neural networks (RNNs), and adopt three popular deep learning tasks: image classification for MNIST, for CIFAR-10 (Krizhevsky, 2009), and sentiment classification for IMDB movie review dataset (Maas et al., 2011).

We use ResNet (He et al., 2015) as the CNN student model and Long-Short-Term-Memory network (Hochreiter & Schmidhuber, 1997) as the RNN student model. Adam (Kingma & Ba, 2014) is used to train the MLP and RNN student models and Momentum-SGD (Sutskever et al., 2013) is used for the CNN student model. We guarantee that the final performance of each student model without teaching matches with previous public reported results. Please refer to Appendix Subsection 8.1 for more details about student models/tasks setup.

#### 5.1.2 DIFFERENT TEACHING STRATEGIES

- *NoTeach*. It means training the student model without any teaching strategy, i.e, the conventional machine learning process.

- *Self-Paced Learning (SPL)* (Kumar et al., 2010). It refers to teaching by the *hardness* of data, as reflected by loss value. Mathematically speaking, those training data $d$ satisfying loss value $l(d) > \eta$ will be filtered out, where the threshold $\eta$ grows from smaller to larger during the training process. To improve the robustness of SPL, following the widely used trick in common SPL implementation (Jiang et al., 2014b), we filter training data using its loss rank in one mini-batch rather than the absolute loss value: we filter data instances with top $K$ largest training loss values within a $M$-sized mini-batch, where $K$ linearly drops from $M - 1$ to 0 during training.

- *Learning to Teach (L2T)*, i.e., the teacher model in L2T framework. The state features $g(s)$ are constructed according to the principles described in Subsection 4.2. We use a three-layer neural network as the policy function $\phi$ for the teacher model. Appendix Subsection 8.2 lists more details of teacher model training.

- *RandTeach*. To conduct comprehensive comparison, for the L2T model we obtained, we record the ratio of filtered data instances per epoch, and then randomly filter data in each epoch according to the logged ratio. In this way we form one more baseline, referred to as RandTeach.

For all teaching strategies, we make sure that *the base neural network model will not be updated until $M$ un-trained, yet selected data instances are accumulated*. That is to guarantee that the convergence speed is only determined by the quality of taught data, not by different model updating frequencies. The model is implemented with Theano and run on one NVIDIA Tesla K40 GPU for each training/testing process.

#### 5.1.3 EVALUATION PROTOCOL

For each teaching strategy in every task, we report the test accuracy with respect to the number of effective training instances. To demonstrate the robustness of L2T, we set different hyper-parameters for both L2T and SPL, and then plot the curve for each hyper-parameter configuration. For L2T, we vary the validation threshold $\tau$ in reward computation. For SPL, we test different speeds to include all the training data during training process. Such a speed is characterized by a pre-defined epoch number $E$, which means all the training data will gradually be included (i.e., $K$ linearly drops from $M - 1$ to 0) among the first $E$ epochs. All the experimental curves reported below are the average results of 5 repeated runs.

To test the generalization ability of the teacher model learnt in the L2T framework, we consider two test settings:

- **Teaching a new student with the same model architecture** (see Subsection 5.2). It refers to train the teacher model using a student model, and then fixed the teacher model to train a new student model with the same architecture. That is, the student model used in the training phase of the teacher model and the student model used in the test phase of the teacher model share the same architecture. The difference between the two student models is that they use different datasets for training. For example, we use the first half of MNIST dataset to train the teacher model for a CNN learner, and apply the teacher to train the same CNN student model on the second half.

- **Teaching a new student with different model architecture** (see Subsection 5.3). Different from the first setting, the two student models in the training and test phases of the teacher model are of different architectures. For example, we use MNIST to train the teacher model for a MLP student, but fix the teacher model to teach a CNN model on CIFAR-10.

## 5.2 TEACHING A NEW STUDENT WITH THE SAME MODEL ARCHITECTURE

In this setting, we have a training set $D_{train}$ and a test set $D_{test}$ for each task. We evenly split the training data $D_{train}$ in each task into two folds: $D_{train}^{teacher}$ and $D_{train}^{student}$. We conduct experiments as follows.

*Step 1*: The first fold $D_{train}^{teacher}$ is used to train the teacher model, with 5% of $D_{train}^{teacher}$ acting as a held-out set $D'_{dev}$ used to compute reward for the teacher model during training.

*Step 2*: After the teacher model is well trained using $D_{train}^{teacher}$, it is fixed to teach and train the student model using the second fold $D_{train}^{student}$. The other teaching strategies listed in Subsection 5.1.2 are also used to teach the student model on $D_{train}^{student}$.

*Step 3*: The student model is tested on the test set $D_{test}$. The accuracy curve of the student model accompany with different teaching strategies on $D_{test}$ is plotted in Fig. 2.

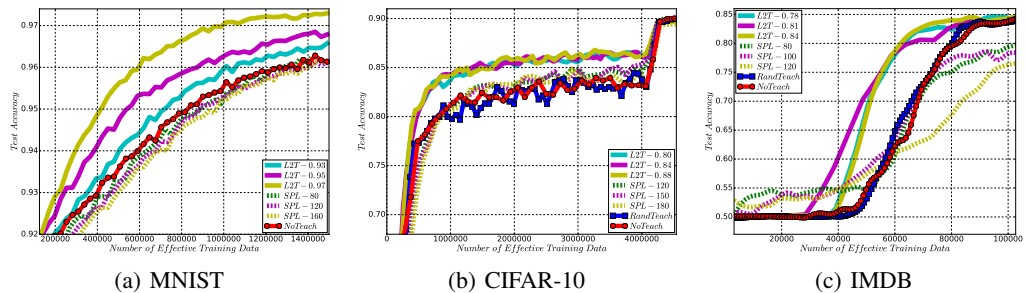

| (a) MNIST | (b) CIFAR-10 | (c) IMDB |

Figure 2: Test accuracy curves of different teaching strategies on MNIST(**a**), CIFAR-10(**b**) and IMDB(**c**). Different hyper-parameter settings are included: The numbers in L2T-$\tau$ and SPL-$E$ respectively represent the two hyper-parameters in L2T and SPL introduced in Subsection 5.1.3.

We can observe that L2T achieves the best convergence speed, significantly better than other teaching strategies in all the three tasks. For example, in MNIST experiments 2(a), L2T achieves a fairly good classification accuracy (e.g, 0.96) with roughly 45% training data of the student model without any data teaching strategy, i.e., the baseline NoTeach. Such a reduction ratio of training data for CIFAR-10 and IMDB is about 50% and 75% respectively. Therefore, we conclude that L2T performs quite well when its learnt teacher model is used to teach a new student model with the same architecture.

### 5.2.1 FILTRATION NUMBER ANALYSIS

To further investigate the learnt teacher model in L2T, in Fig. 3 we show the number of training data it decides to filter in each epoch in Step 2 of the student model training.

There are several interesting observations: (1) For the two image recognition tasks L2T acts quite differently from CL/SPL: as training goes on, more and more data will be filtered. Meanwhile,

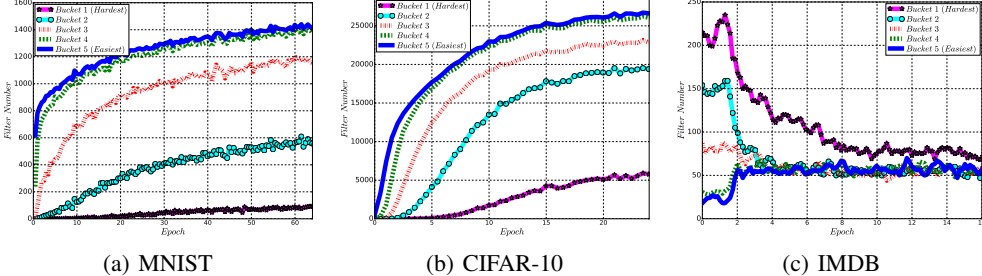

(a) MNIST       (b) CIFAR-10       (c) IMDB

Figure 3: The number of instances filtered by L2T teacher in each training epoch of MNIST**(a)**, CIFAR-10**(b)** and IMDB**(c)**. Different curves denote the number of filtered data corresponding to different hardness levels, as indicated by the ranks of loss on that filtered data instance within its mini-batch. Concretely speaking, we evenly partition all the rank values $\{1, 2, \cdots, M\}$, where $M$ is the batch size, into five buckets. Bucket 1 denotes the hardest data whose loss values are largest among the instances in each mini-batch, while bucket 5 is the easiest.

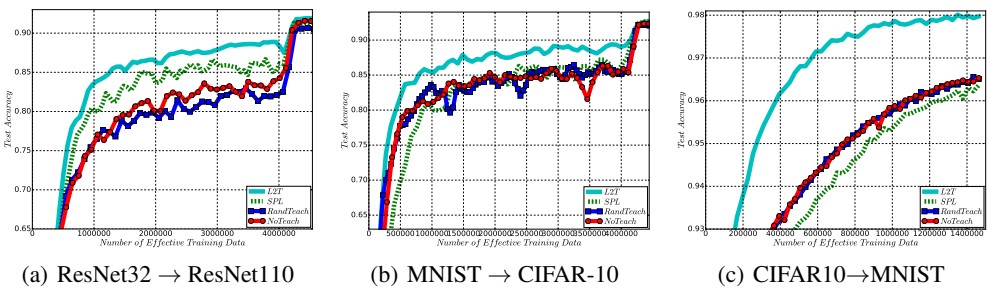

(a) ResNet32 $\rightarrow$ ResNet110    (b) MNIST $\rightarrow$ CIFAR-10    (c) CIFAR10$\rightarrow$MNIST

Figure 4: **(a)**:Apply the teacher trained based on ResNet32 to teach ResNet110 on CIFAR-10. **(b)**: Apply the teacher trained based on MLP for MNIST to train CNN for CIFAR-10. **(c)**:Apply the teacher trained based on CNN for CIFAR-10 to train MLP for MNIST.

hard data (the purple curve) tend to be kept as teaching materials, while easy ones (the green and blue lines) will probably be filtered. Such a result suggests that the student models for MNIST and CIFAR-10 favor harder data as training goes on, whereas those less informative data instances with smaller loss values are comparatively redundant and negligible. (2) In contrast, L2T behaves similarly to CL/SPL for the LSTM student model on IMDB by teaching from easy to hard order. This observation is consistent with previous findings (Zaremba & Sutskever, 2014). Our intuitive explanation is that harder instances on one aspect may affect the initialization of LSTM (Dai & Le, 2015; Wang & Tian, 2016), and on the other aspect are likely to contain noises. Comparatively speaking, MLP and CNN student models are relatively easier to initialize and image data instances contain less noise. Thus, for the two image tasks, the teacher model can provide hard instances to the student model for training from the very beginning, while for the natural language task, the student model needs to start from easy instances. The different teaching behaviors of L2T in image and language tasks demonstrate its adaptivity and applicability to different learning tasks, and seems to suggest the advantage of learning to teach over fixed/heuristic teaching rules.

## 5.3 TEACHING A NEW STUDENT WITH DIFFERENT ARCHITECTURE

In this subsection, we consider more difficult, yet practical scenarios, in which the teacher model is trained through the interaction with a student model and then used to teach another student model with different model architecture.

### 5.3.1 RESTNET32 $\rightarrow$ RESNET110 ON CIFAR-10

The first scenario is using the teacher model trained with ResNet32 as student on the first half of CIFAR-10 training set, to teach a much deeper student model, ResNet110, on the second half

of CIFAR-10 training set. The accuracy curve on the test set is shown in Fig. 4(a). Apparently, L2T effectively collects the knowledge in teaching the student with smaller model, and successfully transfers it to the student with much bigger model capacity.

### 5.3.2 MLP on MNIST ↔ CNN on CIFAR-10

The second scenario is even more aggressive: We first train the teacher model based on the interaction with a MLP student model using the MNIST dataset, and then apply it to teach a ResNet32 student model on the CIFAR-10 dataset. The accuracy curve of the ResNet32 model on the CIAR-10 test set is shown in Fig. 4(b). Similarly, we conduct experiments in the reverse direction, and the results are shown in Fig. 4(c). Again, L2T succeeds in such difficult scenarios, demonstrating its powerful generalization ability. In particular, the teacher model trained on CIFAR-10 significantly boosts the convergence of the MLP student model trained with MNIST (show in Fig. 4(c)).

### 5.3.3 Wall-Clock Time Analysis

Different from previous curves showing the performance w.r.t. the number of effective training data, we in Fig. 5 show the learning curves of training a ResNet32 model on CIFAR-10 using different teaching strategies, but varying with wall-clock time. The teacher model in L2T is trained on MNIST with MLP student models, i.e., the same one with Fig. 4(b). Apparently, even with the process of obtaining all the state features, L2T also achieves training time reduction for the student model through providing high-quality training data.

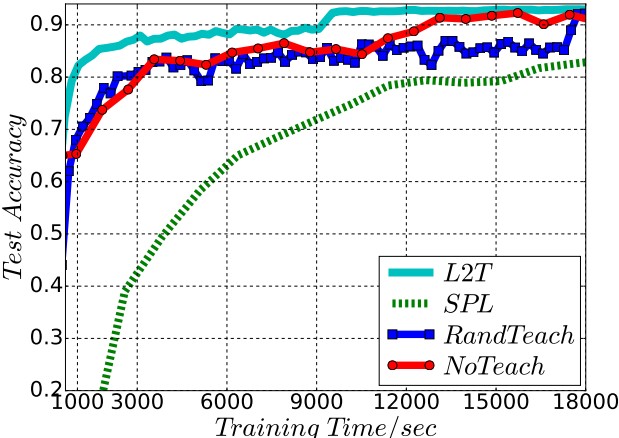

Figure 5: Learning curves w.r.t. wall-clock time of training ResNet32 student model on CIFAR-10 under different teaching strategies.

### 5.4 Teaching for Improving Accuracy

In this subsection, apart from boosting the convergence speed, we show that the teacher model in L2T also helps to improve the final accuracy. The student model is the LSTM network trained on IMDB. We first train the teacher model on half of the training data of IMDB dataset. The terminal reward is defined as the dev set accuracy after the student model is trained for 15 epochs. Then the teacher model is applied to train the student model on the full dataset till its convergence (as indicated by that the dev set accuracy stops to increase). The state features are kept the same as those in previous experiments. The other settings in student model training such as LSTM model sizes are the same as previous work (Dai & Le, 2015) (see subsection 8.1 for more details).

The results are shown in Table 1. Note that the baseline accuracy of *NoTeach* is comparable to the result reported in (Dai & Le, 2015). We can see that L2T achieves better classification accuracy for training LSTM network, surpassing the SPL baseline by more than 0.6 point (with $p\_value < 0.001$).

| Teaching Policy | NoTeach | SPL | L2T |
|---|---|---|---|
| Accuracy | 88.54% | 88.80% | **89.46%** |

Table 1: Accuracy of IMDB sentiment classification using different teaching policies.

## 6 CONCLUSION

Inspired by the education systems in human society, we have proposed the framework of learning to teach, an end-to-end trainable method to automate the teaching process. Comprehensive experiments on several real-world tasks have demonstrated the effectiveness of the framework.

There are many directions to explore for learning to teach in future. First, we have studied the application of L2T to image classification and sentiment analysis. We will study more applications such as machine translation and speech recognition. Second, we have focused on data teaching in this work. As stated in Subsection 3.1, we plan to investigate other teaching problems such as loss function teaching and hypothesis space teaching. Third, we have empirically verified the L2T framework through experiments. It is interesting to build theoretical foundations for learning to teach, such as the consistence and generalization of the teacher model.

## 7 ACKNOWLEDGEMENT

We thank Di He for his helpful suggestions. We thank all the anonymous reviewers' comments to make the paper more comprehensive. The research of Li is partially supported by China National Funds for Distinguished Young Scientists with No. 61625205, Key Research Program of Frontier Sciences, CAS, No. QYZDY-SSW-JSC002, NSFC with No. 61520106007, 61751211, NSF ECCS-1247944, and NSF CNS 1526638.

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

## 8 APPENDIX

### 8.1 TASKS AND STUDENTS SETUP

#### 8.1.1 SETTINGS FOR TRAINING MLP ON MNIST

MNIST dataset consists of $60k$ training and $10k$ testing images of handwritten digits from 10 categories (i.e., $0, \cdots, 9$). SGD with momentum is used to perform MLP model training, and mini-batch size is set as 20. The base model is a three-layer feedforward neural network with $784/500/10$ neurons in its input/hidden/output layers. $tanh$ acts as the activation function for the hidden layer. Cross-entropy loss is used for training.

### 8.1.2 SETTINGS FOR TRAINING CNN ON CIFAR-10

Cifar10 is a widely used dataset for image classification, which contains $60k$ RGB images of size $32 \times 32$ categorized into 10 classes. The dataset is partitioned into a training set with $50k$ images and a test set with $10k$ images. Furthermore, data augmentation is applied to every training image, with padding 4 pixels to each side and randomly sampling a $32 \times 32$ crop. ResNet (He et al., 2015), a well-known effective CNN model for image recognition, is adopted to perform classification on CIFAR-10. Concretely speaking, we use ResNet32 and ResNet110 models, respectively containing 32 and 110 layers. The code is based on a public Lasagne implementation [3]. The mini-batch size is set as $M = 128$ and Momentum-SGD Sutskever et al. (2013) is used as the optimization algorithm. Following the learning rate scheduling strategy in the original paper (He et al., 2015), we set the initial learning rate as $0.1$ and multiply it by a factor of $0.1$ after the $32k$-th and $48k$-th model update. Training in this way the test accuracy reaches about $92.4\%$ and $93.2\%$, respectively for ResNet32 and ResNet110.

### 8.1.3 SETTINGS FOR TRAINING LSTM ON IMDB

IMDB [4] is a binary sentiment classification dataset consisting of $50k$ movie review comments with positive/negative sentiment labels (Maas et al., 2011), which are evenly separated (i.e., $25k/25k$) as train/test set. The sentences in IMDB dataset are significantly long, with average word token number as $281$. Top $10k$ most frequent words are selected as the dictionary while the others are replaced with a special token UNK. We apply LSTM (Hochreiter & Schmidhuber, 1997) RNN to each sentence, taking randomly initialized word embedding vectors as input, and the last hidden state of LSTM is fed into a logistic regression classifier to predict the sentiment label (Dai & Le, 2015). The size of word embedding in RNN is $256$, the size of hidden state of RNN is $512$, and the mini-batch size is set as $M = 16$. Adam (Kingma & Ba, 2014) is used to perform LSTM model training with early stopping based on validation set accuracy. The test accuracy is roughly $88.5\%$, matching the public result in previous work (Dai & Le, 2015).

## 8.2 DETAILS FOR THE TEACHER MODELS IN L2T

In L2T, we use a three-layer neural network, with layer sizes $d \times 12 \times 1$, as the teacher model $\phi_\theta(a|s)$. $d$ is the dimension of $g(s)$ and $tanh$ is the activation function for the middle layer. All the weight values in this network are uniformly initialized between $(-0.01, 0.01)$. The bias terms are all set as $0$ except for the bias in the last-layer which is initialized as $2$, with the goal of not filtering too much data in the early age. Adam (Kingma & Ba, 2014) is leveraged to optimize the policy. To reduce estimation variance, a moving average of the historical reward values in previous episodes is set as a reward baseline for the current episode (Weaver & Tao, 2001). We train the teacher model till convergence, i.e., the terminal reward $r_T$ stops improving for several episodes.

## 8.3 STATE FEATURE ANALYSIS

In this section, we give a detailed list for all the features used to construct the state feature vector $g(s)$ (Subsection 8.3.1), as well as their different importance in making a qualified L2T policy (Subsection 8.3.2).

### 8.3.1 STATE FEATURES $g(s)$

Corresponding to the feature description in Section 4.2 of the paper, we list details of the aforementioned three categories of the features:

- Data features, mainly containing the label information of the training data. For all the three tasks, we use $1$ *of* $|Y|$ representations to characterize the label. Additionally, the sequence length (i.e., word token number), divided by a pre-define maximum token number $500$ and truncated to maximum value $1.0$ if exceeded, is set as an additional data feature for IMDB dataset.

---

[3] https://github.com/Lasagne/Recipes/blob/master/papers/deep_residual_learning/Deep_Residual_Learning_CIFAR-10.py

[4] http://ai.stanford.edu/~amaas/data/sentiment/

- Model features. We use three signals to represent the status of current model $\mathcal{W}_t$: 1) current iteration number; 2) the averaged training loss over past iterations; 3) the best validation loss so far. All the three signals are respectively divided by pre-defined maximum number to constrain their values in the interval $[0, 1]$.

- The combined features. Three parts of signals are used in our classification tasks: 1) the predicted probabilities of each class; 2) the loss value on that data, i.e, $-\log P_y$, which appears frequently in self-paced learning algorithms (Kumar et al., 2010; Jiang et al., 2014a; Sachan & Xing, 2016); 3) the margin value on the training instance $(x, y)$, defined as $P(y|x) - \max_{y' \neq y} P(y'|x)$ (Cortes et al., 2013). For the loss and margin features, to improve stability, we use their (normalized) ranks in the mini-batch, rather than the original values.

Based on the above designs, the dimensions of the state feature vector $g(s)$ for the three tasks are respectively: a) $25 = 10$ (Data features) $+3$ (Model features)$+12$ (Combined features) for MNIST; b) $25 = 10 + 3 + 12$ for CIFAR-10; c) $10 = 3$ (1 of $|Y| = 2$ representation + sequence length) $+3$ (Model features)$+ 4$ (Combined features) for IMDB. The feature vector $g(s)$ is further normalized to satisfy $||g(s)||_2 = 1$.

### 8.3.2 IMPORTANCE ANALYSIS FOR DIFFERENT FEATURES

To better understand how different features play effects in constructing a good L2T policy, we conduct a systematic studies on the importance of different features. Concretely speaking, for all the three categories of features, we respectively remove each of them, and use the remaining two parts as state features $g(s)$ to re-train/test the L2T policy. The base task is training MLP on MNIST dataset since it takes shortest time among all the three tasks in our experiments. The experimental results are shown in Fig. 6.

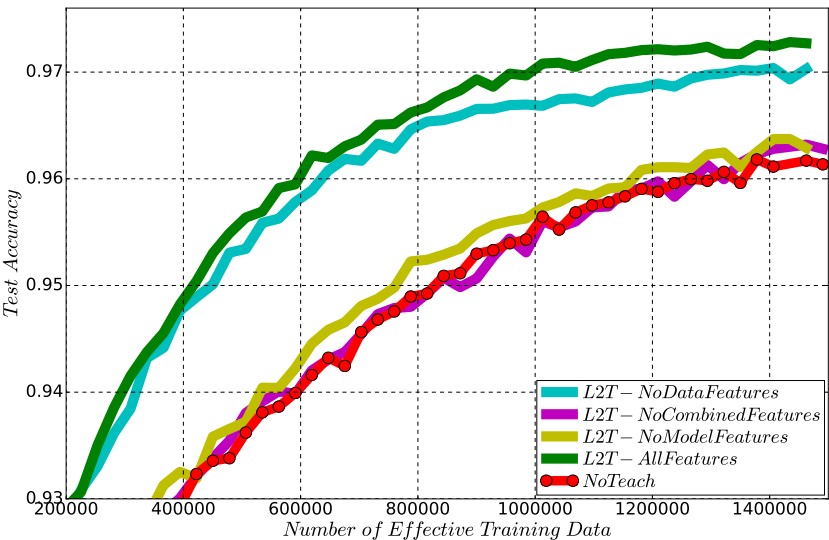

Figure 6: Feature analysis for L2T on MNIST dataset. The learning curves of NoTeach, and L2T with all the three parts of features remained, are also included.

We have the following observations from Fig. 6.

- The model features and the combined features are critical to the success of L2T policy, as shown by the poor convergence when either of the two are removed. Actually without any category of the two subset of features, the performance of SGD with L2T decreases to that of SGD without data scheduling.

- The data features are relatively less important to L2T. By removing the data features, i.e., the label information of the data, the performance of SGD with L2T drops but not by much.

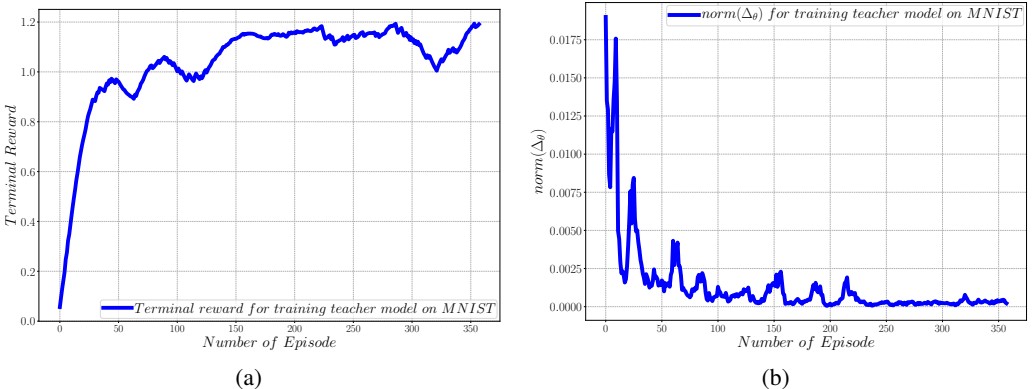

(a)  (b)

Figure 7: **(a)**:The reward curves in each episode of teacher model training. **(b)**: The L2 norm of teacher model weight changes ($\Delta_\theta$) in each episode of teacher model training.

## 8.4 THE CONVERGENCE OF TEACHER MODEL TRAINING

In this subsection, we show the convergence property of training teacher model in L2T. Similar to 8.3.2, we investigate the training of the teacher model used to supervise the MLP on MNIST as the student. In Fig.7(a), we plot the terminal reward (i.e., $r_T = -\log(i_\tau/T')$ in 4.2) in each episode of teacher model training. In Fig.7(b), we plot the $L2$ norm of the teacher model parameter updates in each episode(i.e., $\Delta_\theta(t) = \theta(t+1) - \theta(t)$ for each episode $t$). From both figures, it can be seen that the teacher model trained after 50 episodes is ready to be deployed since the reward is much larger than that of scratch (shown in Fig. 7(a)) and the model variation is small afterwards (shown in Fig. 7(b)).

