# OpenReview forum: "Learning to Teach"
_ICLR.cc/2018/Conference — Accept (Poster)_

### Official Review · AnonReviewer2 · 2017-11-27
**excellent article, though minor revisions to the previous work would strengthen it**

**Rating:** 9
**Confidence:** 3

**Review:**

The authors define a deep learning model composed of four components:  a student model, a teacher model, a loss function, and a data set. The student model is a deep learning model (MLP, CNN, and RNN were used in the paper). The teacher model learns via reinforcement learning which items to include in each minibatch of the data set. The student model learns according to a standard stochastic gradient descent technique (Adam for MLP and CNN, Momentum-SGD for RNN), appropriate to the data set (and loss function), but only uses the data items of the minibatch chosen by teacher model. They evaluate that their method can learn to provide learning items in an efficient manner in two situations: (1) the same student model-type on a different part of the same data set, and (2) adapt the teaching model to teach a new model-type for a different data set. In both circumstances, they demonstrate the efficacy of their technique and that it performs better than other reasonable baseline techniques: self-paced learning, no teaching, and a filter created by randomly reordering the data items filtered out from a teaching model.

This is an extremely impressive manuscript and likely to be of great interest to many researchers in the ICLR community. The research itself seems fine, but there are some issues with the discussion of previous work. Most of my comments focuses on this.

The authors write that artificial intelligence has mostly overlooked the role of teaching, but this claim is incorrect. There is a long history of research on teaching in artificial intelligence. Two literatures of note are intelligent tutoring and machine teaching in the computational learnability literature. A good historical hook to intelligent tutoring is Anderson, J. R., Boyle, C. F., & Reiser, B. J. (1985). Intelligent tutoring systems. Science, 228. 456-462. The literature is still healthy today. One offshoot of it has its own society with conferences and a journal devoted to it (The International Artificial intelligence in Education Society: http://iaied.org/about/).

For the computational learnability literature, complexity analysis for teaching has a subliterature devoted to it (analogous to the learning literature). Here is a hook into that literature: Goldman, S., & Kerns. M. (1995). On the complexity of teaching. Journal of Computer and Systems Sciences, 50(1), 20-31.

One last related literature is pedagogical teaching from computational cognitive science. This one is a more recent development. Here are two articles, one that provides a long and thorough discussion that is a definitive start to the literature, and another that is most relevant to the current paper, on applying pedagogical teaching to inverse reinforcement learning (a talk at NIPS 2016).

Shafto, P., Goodman, N. D., & Griffiths, T. L. (2014). A rational account of pedagogical reasoning: Teaching by, and learning from, examples. Cognitive Psychology, 71, 55-89.

Ho, M. K., Littman, M., MacGlashan, J., Cushman, F., & Austerweil, J. L. (NIPS 2016).

I hope all of this makes it clear to the authors that it is inappropriate to claim that artificial intelligence has “largely overlooked” or “largely neglected”.

One other paper of note given that the authors train a MLP is an optimal teaching analysis of a perceptron: (Zhang, Ohannessian, Sen, Alfeld, & Zhu, 2016; NIPS).

---

> ### Author Response · Authors · 2017-12-13
> **Thanks for offering the rich literature**
>
> We thank you very much for letting us know so many literatures about teaching, both from the view of cognitive science and computational learnability. All these works are related with ours and we have included them to make our manuscript more comprehensive. We have also removed some improper statements.
> Here are several preliminarily discussions on the difference of the listed literature and L2T:
>
> 1.	As we pointed in the manuscript, some of the works hold strong assumption of an oracle existence for the teacher model, such as (Zhang, Ohannessian, Sen, Alfeld, & Zhu, 2016; NIPS) and (Goldman, S., & Kerns. M. On the complexity of teaching. Journal of Computer and Systems Sciences, 1995).
>
> 2.	The literature of pedagogical teaching, especially its application to IRL (Ho, M. K., Littman, M., MacGlashan, J., Cushman, F., & Austerweil, J. L. ,NIPS 2016) is much closer to our setting in that the teacher adjusts its behavior in order to facilitate student learning, by communicating with student (i.e., showing not doing).  Apart from some differences in experimental setup and application scenarios, the key difference between the two works is in L2T we are using automatically learnt strategies for the teacher model that can be transferred between tasks. Furthermore, applications of pedagogical teaching in IRL implies that the teacher model is still much stronger than student, somehow similar to the oracle existence assumption in point 1 since there is an expert in IRL that gives the (state, action) trajectories based on the optimal policy.

---

### Official Review · AnonReviewer3 · 2017-11-27
**Review of "Learning to Teach"**

**Rating:** 8
**Confidence:** 4

**Review:**

This paper focuses on the problem of "machine teaching", i.e., how to select a good strategy to select training data points to pass to a machine learning algorithm, for faster learning. The proposed approach leverages reinforcement learning by defining the reward as how fast the learner learns, and use policy gradient to update the teacher parameters. I find the definition of the "state" in this case very interesting. The experimental results seem to show that such a learned teacher strategy makes machine learning algorithms learn faster.

Overall I think that this paper is decent. The angle the authors took is interesting (essentially replacing one level of the bi-level optimization problem in machine teaching works with a reinforcement learning setup). The problem formulation is mostly reasonable, and the evaluation seems quite convincing. The paper is well-written: I enjoyed the mathematical formulation (Section 3). The authors did a good job of using different experiments (filtration number analysis, and teaching both the same architecture and a different architecture) to intuitively explain what their method actually does.

At the same time, though, I see several important issues that need to be addressed if this paper is to be accepted. Details below.

1. As much as I enjoyed reading Section 3, it is very redundant. In some cases it is good to outline a powerful and generic framework (like the authors did here with defining "teaching" in a very broad sense, including selecting good loss functions and hypothesis spaces) and then explain that the current work focuses on one aspect (selecting training data points). However, I do not see it being the case here. In my opinion, selecting good loss functions and hypothesis spaces are much harder problems than data teaching - except maybe when one use a pre-defined set of possible loss functions and select from it. But that is not very interesting (if you can propose new loss functions, that would be way cooler). I also do not see how to define an intuitive set of "states" in that case. Therefore, I think this section should be shortened. I also think that the authors should not discuss the general framework and rather focus on "data teaching", which is the only focus of the current paper. The abstract and introduction should also be modified accordingly to more honestly reflect the current contributions.
2. The authors should do a better job at explaining the details of the state definition, especially the student model features and the combination of data and current learner model.
3. There is only one definition of the reward - related to batch number when the accuracy first exceeds a threshold. Is accuracy stable, can it drop back down below the threshold in the next epoch? The accuracy on a held-out test set is not guaranteed to be monotonically increasing, right? Is this a problem in practice (it seems to happen on your curves)? What about other potential reward definitions? And what would they potentially lead to?
4. Experimental results are averaged over 5 repeated runs - a bit too small in my opinion.
5. Can the authors show convergence of the teacher parameter \theta? I think it is important to see how fast the teacher model converges, too.
6. In some of your experiments, every training method converges to the same accuracy after enough training (Fig.2b), while in others, not quite (Fig. 2a and 2c). Why is this the case? Does it mean that you have not run enough iterations for the baseline methods? My intuition is that if the learner algorithm is convex, then ultimately they will all get to the same accuracy level, so the task is just to get there quicker. I understand that since the learner algorithm is an NN, this is not the case - but more explanation is necessary here - does your method also reduces the empirical possibility to get stuck in local minima?
7. More explanation is needed towards Fig.4c. In this case, using a teacher model trained on a harder task (CIFAR10) leads to much improved student training on a simpler task (MNIST). Why?
8. Although in terms of "effective training data points" the proposed method outperforms the other methods, in terms of time (Fig.5) the difference between it and say, NoTeach, is not that significant (especially at very high desired accuracy). More explanation needed here.

Read the rebuttal and revision and slightly increased my rating.

---

> ### Author Response · Authors · 2017-12-13
> **More reward setup and analysis are provided, with some clarifications**
>
> Thank you very much for your detailed comments and suggestions. Here are several of our responses:
>
> (1)	Thank you for pointing the writing issues in section 3, as well as the suggestions on loss function design. Yes, in current manuscript we are demonstrating the effects of L2T in the scenario of data selection. Meanwhile we are exploring the potential of L2T for designing better loss functions for neural machine translation.  We will update the paper once we get meaningful results.
>
> (2)	For the details of state definition, please refer to section 7.3 of the Appendix. In 7.3.1, we list the feature details and in 7.3.2 we conduct the ablation study of different features.
>
> (3)	The current reward function is designed for guiding the reinforcement learning process to achieve better convergence. It is true that we cannot guarantee the monotonicity of reward during the training of the teacher model. However, the general increasing trend can be observed.  Please refer to our new Appendix 7.4 for more details (as well as teacher model parameter convergence). Furthermore, it is for sure that we can use other rewards such as the final accuracy on a held-out dev set when our aim is to improve the accuracy. We have included in the new version (subsection 5.4) and please have a check.
>
> (4)	For different final accuracies (your point 6), your intuition is right and it is simply an issue of figure drawing: for example, in Fig. 2(c), we will have to use a very wide figure if we want to draw the entire curves of SPL, given that SPL converges very slowly.  In terms of final accuracy, different teaching strategies are roughly the same, with SPL on IMDB a little bit higher. On the other hand, in L2T if we use a new reward that indicates the final accuracy, not the convergence speed, we can achieve better final accuracy. All these facts are reflected in our new subsection 5.4.
>
> (5)	For Fig. 4(c), our explanations are as follows: first, CIFAR10 (colored images, diverse objects) contains more information than MNIST (black/white images, only digits objects); second, the ResNet model on CIFAR10 is much more powerful than the simple MLP model on MNIST. These two factors make the teacher model trained on CIFAR10 more precise and useful. Intuitively speaking, simple tasks are easy to solve and learning a teacher model becomes not critically necessary (and as a result, the learning of the teacher model on simple tasks might not be sufficient). In contrast, harder tasks are not easy to solve without the teacher model, and therefore the teacher model could be learned more sufficiently with more useful feedback signals.
>
> (6)	For the wall-clock time analysis in Fig.5, our current implementation of L2T is not optimal and the comparison is a little unfair to L2T. Actually, due to the limitation of Theano, since the computational graph is pre-built, we cannot directly implement the idea of L2T and will have to use some redundant computations. Specifically, our current implementation of L2T contains two rounds of feedforward (ff) process for the selected data via our teacher model. Ideally, we only need one round of feedforward, since the loss values out of this feedforward process do not only constitute the state features but also directly derive the gradients for backpropagation.  However, given the limitation of Theano, we have to go through another round of ff: after the data is selected by the teacher model, they will re-enter the pre-built computational graph, leading to another round of (wasteful) feedforward. It is a pity that we have to bear such an inefficient implementation at this moment, and we plan to try more flexible deep learning toolkits in the future. So in theory, L2T will be significantly more efficient in terms of wall-clock time, although the current experiments could not show that significance.

---

### Official Review · AnonReviewer1 · 2017-11-30
**Review for Learning to Teach**

**Rating:** 5
**Confidence:** 4

**Review:**

This paper suggests a "learning to teach" framework. Following a similar intuition as self-paced learning and curriculum learning, the authors suggest to learn a teaching strategy,  corresponding to choices over the data presented to the learner (and potentially other decisions  about the learner, such as the  algorithm used).   The problem is framed as RL problem, where the state space corresponds to learning configurations, and teacher actions change the state.  Supervision is obtained by observing the learner's performance.

I found it very difficult to understand the evaluation.
First, there is quite a bit of recent work on learning to teach and curriculum learning.  It would be helpful if there are comparisons to these models, and use similar datasets.  It's not clear if an evaluation on the MNIST data set is particularly meaningful.   The implementation of SPL seems to hurt performance in some cases (slower convergence on the IMDB dataset), can you explain it?  In other text learning task (e.g., [1]) SPL showed improved performance.   The results over the IMDB dataset in the original paper [2] are higher than the ones reported here, using a simple model (BoW).
Second, in non-convex problems, one can expect curriculum learning approaches to also perform better, not just converge faster.  This aspect is not really discussed.  Finally,  I'm not sure I understand the X axis in Figure 2, the (effective) number of examples is much higher than the size of the dataset. Does it indicate the number of  iterations over the same dataset?

I would also like to see some analysis of what's actually being learned by the teacher. Some qualitative analysis, or even feature ablation study would be helpful.

[1] Easy Questions First? A Case Study on Curriculum Learning for Question Answering. Sachan et-al.
[2] Learning Word Vectors for Sentiment Analysis. Maas et-al.

---

> ### Author Response · Authors · 2017-12-13
> **The updated response for clearer clarification**
>
> Thank you for your review comments. We would like to make several points for the sake of clarification:
>
>  (1) As the usage of MNIST, although we have reported the convergence results on this dataset (Fig. 2(a)), what we would like to emphasize is that the learnt teaching policy can be transferred across datasets and model structures: from simple dataset (e.g., MNIST) to relatively difficult dataset (e.g., CIFAR10, in Fig. 4(b)), and from simple models (e.g., MLP) to advance models (e.g., ResNet, in Fig. 4(b) ), for the sake of **improving convergence**. We agree that in terms of **improving final accuracy**, it is meaningful to compare with some recent literature on curriculum learning such as [1]. We leave it as future work and has not reported it currently because: a) To show that L2T also works for improving accuracy, we have provided a new experiment on improving IMDB classification accuracy in subsection 5.4 where SPL has gain over baseline; b) we have not found public code implementation of [1]. Given that there are quite a few domain specific heuristics in [1] and we have not worked on QA tasks before, it takes more efforts to exactly reproduce their results.
>
> (2) For SPL, we guarantee that our implementation is correct (details reported in section 5.1.2). We also noticed that SPL performs not very well on the IMDB dataset, and our explanation is that SPL may not be very robust when working with LSTM (actually, by analyzing different filtered data patterns in Fig. 3(a-c), we can observe this kind of incompatibility to some extent).  However, we do not think our experimental finding is inconsistent with [2]. Please note that the better number (88.89%) on IMDB in [2] was obtained with additional unlabeled data. With labeled data only, their number is 88.33%, which is even worse than our implementation (88.5% as reported in Appendix 7.1.3).  Furthermore, we never aim to propose a model to achieve state-of-the-art numbers on this task (i.e., IMDB). Our goal is to demonstrate the generality of L2T in achieving better convergence for various neural networks training tasks, and thus we take IMDB as a testbed for LSTM networks. Another important note: in the beginning of section 5.2, we have pointed out that we train our teacher model on the first half of the data and apply it to training the student model with the second half. This is to guarantee that we do not mix the data in the training and testing phases of the teacher model. As a result, the curves on IMDB in Fig 2(c) were all obtained using half of the standard training data, and it is understandable that the corresponding results are a little worse (about 85% accuracy).
>
> (3) We agree that sometimes better accuracy could be the goal of L2T. In this case, we just need to change the reward function from “the batch number when accuracy first exceeds a threshold” to “the final accuracy on a held-out set”. Please check our new subsection 5.4 for more results (also mentioned in our point (1)).
>
> (4) For Fig. 2, ‘effective training instances’ in the X axis include the instances used for training the model, which may contain multiple instances of the same data sample.  It is possible that this number will exceed the size of the training dataset because one usually needs to sweep the dataset for multiple epochs during the training process.
>
> (5) For your last point, i.e., ‘some qualitative analysis, or even feature ablation study would be helpful’, we actually have already done this in our paper – please refer to Section 7.3 (State Feature Analysis’) of the Appendix. In that section, we do not only list the detailed features, but also conduct comprehensive ablation study on the effects of different features.

---

> ### Author Response · Authors · 2017-12-29
> **Add a new experiment on improving accuracy**
>
> Dear Reviewer,
>
> We've added a new experiment towards verifying the effectiveness of L2T in terms of improving accuracy (see subsection 5.4). We hope that the new results, together with our previous rebuttal for clarifications, can remove several of your concerns. Thanks.
>
> Best,
> The Authors

---

> ### Author Response · Authors · 2018-01-15
> **Any Further Questions or Concerns?**
>
> Dear Reviewer,
>
> Do you have any further questions/concerns towards our new paper/rebuttal?
>
> Best Regards,
> The Authors

---

### Public Comment · (anonymous) · 2017-12-13
**Very interesting work. Some questions about baselines**

Thank you for a very interesting paper! It was very clearly written and I especially enjoyed the thorough discussion of related work.

I was wondering if the authors tried a simpler (and potentially better) baselines than SPL? For example, some of the baselines considered in Graves et al. [1] should be very trivial to implement: prediction gain (I think this is very similar to SPL),  gradient prediction gain, etc. Relatedly, any thoughts on why SPL doesn't work at all?

From Appendix 7.3.2 (I recommend upgrading this to the main part of the paper, by the way, since I found it to be one of the most illuminating sections) it seems clear that most of the work done is based on the combined features, which look very similar to heuristics considered previously.

[1] https://arxiv.org/pdf/1704.03003.pdf

---

> ### Author Response · Authors · 2018-01-04
> **About SPL and some baselines**
>
> Thanks very much for your interests and comments to our work!
>
> (1)	We have not claimed, and do not think that our experiments show that ‘SPL doesn’t work at all’. First, in terms of convergence speed, SPL slightly outperform baseline (NoTeach) on CIFAR-10 (Fig.2. (b)). Furthermore, it also works for the initialization process of LSTM on IMDB (Fig. 2(c)), but after the initiation process, the static pattern (gradually include the data) of SPL makes the data usage highly inefficient; Second, in terms of final accuracy, SPL is better than NoTeach, please refer to our new subsection 5.4.
>
> (2)	Thank you for the suggestion of changing the position of Appendix 7.3.2. The figure shows that most of the work done is based on **both combined features and model features**, not only combined features. Furthermore, the key difference of L2T and ‘ heuristics considered previously’ is the weights of these features in L2T are automatically learnt and transferred.
>
> (3)	We appreciate the suggestions of trying other signals in curriculum learning such as those used in [1]. As you say, some of the signals (prediction gain) are essentially similar to the loss values used in SPL, and they can be also included into our L2T feature space.

---

### Author Response · Authors · 2017-12-13
**Thank you very much for the helpful reviews**

Dear Reviewers,

We thank all your constructive review comments, which definitely help the improvement of the paper! We are providing the first-round feedback for clarifications, without revisions of the current manuscript. After that, we will provide a new round of paper in the next several weeks.

Best

---

### Author Response · Authors · 2018-01-04
**The latest paper version**

Dear reviewers,

We modified our paper in that:

1) A new subsection 5.4 is added to show the performance of L2T in improving accuracy, as well as other rewards setup, as suggested by reviewer 1 and 3;

2) A new subsection Appendix 7.4 is added to show the convergence property of teacher model training, as suggested by reviewer 3;

3) Added the missed references in section 2, and modified some inappropriate statements as suggested by reviewer 2.

We also updated our initial response to all of you, for the sake of clearer clarifications and looping in the latest manuscript changes. We hope all these can make our paper more comprehensive and remove your corresponding concerns. Thanks!

---

### Public Comment · (anonymous) · 2018-01-23
**Some questions about the details of the paper**

Dear the authors,

I really like this work, it's great and insightful.
I have some questions about the implementation of the model.

1). Could you provide the exact setting of hyper parameter: T' (the maximum iteration number)?

2). In section 5.1.2, you mentioned that "the base neural network model will not be updated
until M un-trained, yet selected data instances are accumulated.". So, when the teacher model selected more than M samples (say N ) for training, you just drop the N-M samples?

3). I am not pretty sure how you train the teacher network. Based on my understanding, the teacher network scan the training data and selects data for training the student network, and once it has collected $M$ data points, then update the student network using those $M$ data points. After the update, we test the student network on Dev set, and check if it reaches the expected accuracy.  And once it reaches the expected accuracy or exceeds the max iteration number, the reward is either 0 or $-log i_{\tau}/T'$. Then, we reinitialize the student network randomly for the next round? Is the mini-batch index i_{\tau} is the number of batches fed into the student network?

<del>4). Could you provide the number of images (for both cifar-10 and mnist) in D^{teacher}_{train} and D_{train}? How do you split the datasets, just randomly? </del> (Solved)

5). Could you provide the exact time for training the teacher network, e.g. using one NVIDIA Tesla K40 GPU, on CIFAR-10

6). The results on Figure 6 is quite surprising to me, I don't understand why the model features (only the iteration number, averaged training loss, and best validation loss) are so important. In my view, those features are not so informative as data features, right? Quite confused..

7). How did you schedule the learning rate of training the teacher network, and the learning of training the student network when training the teacher network? How did you normalize the three signals in model features to be in the interval [0,1] (i.e. what's the pre-defined maximum number to constrain their values in the interval [0,1])? Is the best validation loss is computed on the Dev set?

Thanks for your excellent work, I really enjoy reading it!!

---

### Decision · Program_Chairs · 2018-01-29
**ICLR 2018 Conference Acceptance Decision**

**Decision:**

Accept (Poster)

**Comment:**

The paper addresses the problem of learning a teacher model which selects the training samples for the next mini-batch used by the student model. The proposed solution is to learn the teacher model using policy gradient. It is an interesting training setting, and the evaluation demonstrates that the method outperforms the baseline. However, it remains unclear how the method would scale to larger datasets, e.g. ImageNet. I would strongly encourage the authors to extend their evaluation to larger datasets and state-of-the-art models, as well as include better baselines, e.g. from Graves et al.